# Comprehensive Evaluation of the Bioactive Composition and Neuroprotective and Antimicrobial Properties of Vacuum-Dried Broccoli (*Brassica oleracea* var. *italica*) Powder and Its Antioxidants

**DOI:** 10.3390/molecules28020766

**Published:** 2023-01-12

**Authors:** Antonio Vega-Galvez, Elsa Uribe, Alexis Pasten, Javiera Camus, Luis S. Gomez-Perez, Nicol Mejias, René L. Vidal, Felipe Grunenwald, Lorgio E. Aguilera, Gabriela Valenzuela-Barra

**Affiliations:** 1Food Engineering Department, Faculty of Engineering, Universidad de La Serena, La Serena 1700000, Chile; 2Instituto de Investigación Multidisciplinario en Ciencias y Tecnología, Universidad de La Serena, La Serena 1700000, Chile; 3Center for Integrative Biology, Facultad de Ciencias, Universidad Mayor, Santiago 8380000, Chile; 4Biomedical Neuroscience Institute, University of Chile, Santiago 8380000, Chile; 5Geroscience Center for Brain Health and Metabolism, Santiago 8380000, Chile; 6Departamento de Biología, Facultad de Ciencias, Universidad de La Serena, La Serena 1700000, Chile; 7Laboratorio de Productos Naturales, Facultad de Ciencias Químicas y Farmacéuticas, Universidad de Chile, Santiago 8380000, Chile

**Keywords:** biological activity, drying, natural antimicrobial, neurodegenerative diseases

## Abstract

In this study, vacuum drying (VD) was employed as an approach to protect the bioactive components of and produce dried broccoli powders with a high biological activity. To achieve these goals, the effects of temperature (at the five levels of 50, 60, 70, 80 and 90 °C) and constant vacuum pressure (10 kPa) were evaluated. The results show that, with the increasing temperature, the drying time decreased. Based on the statistical tests, the Brunauer–Emmett–Teller (BET) model was found to fit well to sorption isotherms, whereas the Midilli and Kucuk model fit well to the drying kinetics. VD has a significant impact on several proximate composition values. As compared with the fresh sample, VD significantly reduced the total phenol, flavonoid and glucosinolate contents. However, it was shown that VD at higher temperatures (80 and 90 °C) contributed to a better antioxidant potential of broccoli powder. In contrast, 50 °C led to a better antimicrobial and neuroprotective effects, presumably due to the formation of isothiocyanate (ITC). Overall, this study demonstrates that VD is a promising technique for the development of extracts from broccoli powders that could be used as natural preservatives or as a neuroprotective agent.

## 1. Introduction

*Brassica oleracea* var. *italica*, commonly known as broccoli, is mainly valued for its edible parts, i.e., florets and sprouts. Broccoli florets are an abundant source of diverse nutrients, such as essential amino acids, minerals, vitamins and dietary fiber [1,2], as well as natural phytochemicals (e.g., flavonoids, carotenoids, alkaloids, phytosterols chlorophyll, phenolic compounds and glucosinolates) [3,4]. Although intact glucosinolates are considered biologically inactive compounds [5], they are hydrolyzed upon contact with the enzyme myrosinase, which is released when plant tissues are disrupted, to produce to a range of biologically active compounds, including ITCs, thiocyanates, nitriles, epithionitriles and indoles [6]. These breakdown products, especially ITCs, are known to protect against various types of cancers by reducing cancer cell proliferation [5,7,8] and neurodegenerative diseases [9,10], due to their antioxidant, anti-inflammatory and anti-apoptotic properties [11,12]. ITCs have been proven to exert neuroprotective effects in several in vitro and in vivo models of acute and chronic neurodegenerative disease, especially for their peculiar ability to activate the nuclear factor erythroid 2-related factor 2/antioxidant responsive element (Nrf2/ARE) pathway [9]. In addition, ITCs have been recognized as potential candidates for new antimicrobial compounds with a broad spectrum of activity due to their ability of reducing oxygen consumption and depolarizing the mitochondrial membrane in bacterial cells [13]. Due to the featured medicinal advantages of broccoli florets, their consumption has increased over 940% over the last 35 years [14]. In fact, global production of broccoli and cauliflower has increased by about 6 million ton in the last 10 years, with approximately 26.92 million ton produced in 2019 [4]. In Chile, according to data from the National Statistical Institute (INE) of Chile, Horticultural Survey 2019, the average annual growth rate for the last 5 years of the area of planted broccoli grew by 11.1% according to the Chilean government’s Agricultural Research and Policies Office (ODEPA). In order to provide sustainability to the production of this vegetable, it seems essential to process the significant volume of broccoli to be converted into value-added products (e.g., powder) directed towards the field of development and fortification of complex matrices, such as processed foods [4,14].

Drying is one of the oldest preservation methods utilized for the stabilization of the raw material to prevent microbial growth, deteriorative chemical reactions and degradation reactions caused by enzymes [15]. However, the choice of the suitable drying method is a decision to be taken considering several factors, such as energy consumption, conservation of nutritional value and drying time [16]. In recent years, several novel drying methods have been tested to enhance the quality retention of broccoli as well as to improve drying efficiency. Among these methods, we highlight microwave-assisted hot air-drying [17], microwave vacuum drying [18], air-borne ultrasound-assisted air drying [2] and refractance window drying [19]. Nonetheless, further research is required to use effectively these drying methods in the food industry.

VD is a practical method that has been successfully used for many years in several food industries, taking advantage of low-temperature and oxygen-free conditions [20]. Using this method, drying occurs in a reduced-pressure environment, which lowers the heat required for rapid drying, being more attractive for industrial use. Improving the drying rate and lowering the energy consumption is of great value for the food industry [21].

To our knowledge, studies of this cruciferous vegetable on VD at different temperatures that encompass the mathematical modeling of drying kinetics and its effect on nutritional quality, biocompounds and biological activity have not yet been reported. In fact, only one study has demonstrated that the vacuum environment during VD process at 60 °C can inhibit oxidative reactions and reduce the degradation of the heat-sensitive nutrients of broccoli, such as ascorbic acid, chlorophylls and glucosinolates [18].

In view of the above limitations, the aim of the present study is to evaluate the effect of different VD temperatures (at 50, 60, 70, 80 and 90 °C) on the drying curves and kinetic models of broccoli (*Brassica oleracea* var. *italica*) as well as on bioactive components and their antioxidant, neuroprotective and antimicrobial properties. This study will contribute to a better understanding of VD and provide an alternative method for obtaining broccoli powders of high biological activity.

## 2. Results and Discussion

### 2.1. Drying Kinetics and Mathematical Modeling

The experimental isotherm curves at 60 and 80 °C were evaluated, showing a sigmoidal behavior classified as desorption type II [22]. Figure 1a shows both isotherms; these curves are commonly reported for agri-food products [23]. Desorption curves can be described by zones. In this case, a higher desorption area at water activity ($a_{w}$) > 0.75 was observed, which is related to the water available for reactions and microbial development. Then, there is a central area associated to the multilayer area (0.4 < $a_{w}$ < 0.75), where the molecules of water are less attached to the food structure and the monolayer zone was finally observed with $a_{w}$ < 0.4, which is constituted for bound water molecules [24], considered as unavailable water [25]. The experimental data at 80 °C, especially in the monolayer zone, registered lower moisture values, decreasing the content of bound water fraction. This observation is relevant for equilibrium moisture determination at high temperatures because a faster water diffusion takes place.

The experimental data of both desorption isotherms (60 and 80 °C) were fitted with the BET model to determine the equilibrium moisture ($X_{we}$). The quality fit of the BET model was evaluated statistically by the coefficient of determination (R^2^), the sum of squared errors (SSE) and chi-squared (χ^2^), obtaining average values of 0.87, 0.0255 and 0.0081, respectively. The values of the model parameters at 60 °C were $X_{m}$ = 0.0578 g water/g dry matter (d.m.) and C_BET_ = 17.634 and, at 80 °C, the values were $X_{m}$ = 0.0726 g water/g d.m. and C_BET_ = 8.107.

The BET model data at 60 °C were used to determine the $X_{we}$ values between 50 and 70 °C and the BET model data at 80 °C allowed us to obtain $X_{we}$ at 80 and 90 °C. The calculated values of $X_{we}$ for 50, 60, 70, 80 and 90 °C were 0.0660, 0.0515, 0.0404, 0.0246 and 0.0179 g water/g d.m, respectively, observing a tendency to decrease as the temperature increases.

Figure 1b shows the drying kinetic curves at different temperatures, where the moisture ratio (MR) decreases in time with an exponential curve. This behavior is expected in the drying of broccoli [17,26,27], observing a faster drying rate as the process temperature increases. Seven mathematical models were evaluated to fitting the kinetic experimental data. The Midilli and Kucuk model proved to have the best fit, as shown in Figure 1b, describing in detail the drying process at all the proposed temperatures. Similar findings were also reported by Doymaz and Sahin [26] in broccoli slices.

Table 1 shows the coefficients of each mathematical model evaluated and the quality fit by R^2^, SSE and χ^2^.

The coefficients (*a*, *n*) of the Midilli and Kucuk model are dimensionless, while *k* is (time)*^−n^* and *b* is (time)^−1^. The constant *n* is described as the curvature factor and *k* as the slope factor, being related with the drying rate [28].

The modeling results show a general tendency, where the coefficient *k* increased with the process temperature, and thus the effect of temperature was evaluated using an Arrhenius-type equation (Equation (6)), plotting *ln* (*k*) and 1/T (K^−1^).
(1)$$k=k_{0}e^{\frac{-E_{a}}{RT}}$$

The linear regression allowed us to obtain the value of the pre-exponential constant $k_{0}$ = 4.412 and the activation energy, ($E_{a}$) = 22.78 kJ/mol. The value of $E_{a}$ represents the energy required during the drying process. The variation of the constant *k* with the process temperature can be represented by the following equation: $k=e^{\left[1.4845-\frac{2740}{T\left(K\right)}\right]}$.

### 2.2. Proximate Composition of Fresh and Powdered Broccoli

Broccoli is one of the most common cruciferous vegetables in the world and its great variety in nutrients may contribute to health benefits [4]. The changes in proximate composition of broccoli samples during VD are presented in Table 2. The moisture content of the fresh cruciferous vegetable reached 87.25%. This value was close to that obtained by Campas-Baypoli et al. [29], who demonstrated that the moisture content of broccoli florets was 87.0%. VD; especially at the most intense temperatures, the moisture content was reduced to safe limits, which is reflected in $a_{w}$ values below 0.60 (Table 2). Under these conditions, biological and chemical alteration reactions, catalyzed or facilitated by water, are all inhibited [30].

According to Table 2, raw broccoli contains 1.18 fat, 11.25 ash, 29.91 protein and 11.47 fiber g/100 g d.m.. These values are comparable to those presented for fresh broccoli in a recent review [4]. Our findings show that VD has a significant impact on several proximate composition values of broccoli. For instance, the fat content increased almost four times after drying, regardless of the drying temperature used. An increase in fat content after drying might be due to the breakdown of cell wall and rupture of the underlying epidermal tissue, coagulation of the protein and reduction in oil viscosity, which increases the fat content [31,32]. All dried samples have significantly reduced (*p* ˂ 0.05) the amount of ash compared to the fresh ones. Samples dried at 50 °C obtained the highest ash content, whereas at 80 °C, it was the lowest. This can be explained by the fact that progression of the temperature inside the sample can lead to a weakening of the cell membrane, facilitating water evaporation. Thus, the released water could leach part of the minerals while samples are drying [30]. As far as the protein content is concerned, its values increased significantly after drying, particularly at 50 and 70 °C. Such an increase may be due to a hydrolysis of the thioglucoside linkage catalyzed by myrosinase and, consequently, to the formation of glucose and an unstable aglycone [33]. The formed hydrogen bonds between the carbohydrate and active protein could be responsible for improving dehydration stress, assuring thus the protein stability during the drying process [34]. Concerning the fiber content, a significant decrease (*p* ˂ 0.05) in this parameter was registered in all dried samples compared to the fresh one. This decrease could be attributed to the heat and process time used during VD that could solubilize and degrade some pectic substances in the vegetable matrices.

### 2.3. Bioactive Compounds and Antioxidant Potential of Fresh and Powdered Broccoli

Broccoli contains a variety of non-nutritive compounds, known as phytochemicals or bioactive compounds, and these compounds may have various health benefits. However, they are sensitive to long-time heat during drying and can be easily degraded [18]. The effect of VD on the total phenolic content (TPC), total flavonoid content (TFC) and total glucosinolate content (TGC) of broccoli is depicted in Figure 2. For fresh broccoli, these contents were 5.38 mg gallic acid equivalents (GAE)/g d.m., 13.75 mg quercetin equivalents (QE)/g d.m. and 44.60 µmol sinigrin equivalents (SE)/g d.m., respectively. These values are in the same range than those reported in the literature, where the TPC in broccoli varied from 2.51 to 8.92 mg GAE/g d.m. [1,3,18,35]; TFC varied from 5.4 to 17.5 mg QE/g d.m. [1,36]; and TGC from 19.79 to 73.53 µmol SE/g d.m. [3,18,37]. After drying, the TPC (Figure 2A) was in the range of 3.86–5.01 mg GAE/g d.m., and there was no significant difference in both fresh and dried samples at 80 and 90 °C (*p* > 0.05). Nevertheless, the TPC of dried samples at 50, 60 and 70 °C decreased significantly (*p* < 0.05) by 28.3, 28.3 and 24.9%, respectively, compared to the values of the fresh ones. The loss in TPC may be due to the activation of enzymes, such as polyphenol oxidase and peroxidase, during the drying process at these temperatures, which can interact with some phenolic compounds released from the cell matrix, provoking a strong enzymatic reaction, which leads to the TPC decrease [38]. These reactions, which decrease the concentration of phenolic compounds, do not occur at higher drying temperatures (>80 °C) since fast heating in a short time hampers the activity of oxidative enzymes, protecting phenols from being consumed by those enzymes [39]. Similar results were found in the report of Yilmaz et al. [35], who observed that microwave-dried broccoli at a 54 W/g power intensity level presented a comparable TPC value to that of fresh broccoli and both were superior to convective- and microwave-dried samples at 18 and 36 W/g. Similarly, Cao et al. [2] tested two microwave power levels (500 and 900 W) against air-borne ultrasound-assisted air drying and stated the superiority of the microwave options, this time for broccoli florets. Such results were attributed to a low drying time achieved by a microwave dryer, when compared to other drying conditions and methods.

The TFC of vacuum-dried broccoli ranged from 3.51 to 7.03 mg QE/g d.m., which means 49–74% of TFC was lost during VD (Figure 2B). It has been reported that flavonoids are sensitive to thermal processes and can be easily degraded during drying [20]. The loss of TFC is in line with the results of Xu et al. [18], which revealed that different drying methods may decrease the flavonoids of broccoli.

As shown in Figure 2B, a lower degradation of TFC was found in both the drying of broccoli at 80 and 90 °C, being highly consistent with that of TPC. This might be attributed to the fact that the high drying temperature reduced the time of exposure to heat, which decreased the thermal degradation of TFC [32]. Additionally, high drying temperatures could be responsible for the liberation of some flavonoids that are mainly found in bound form in plant matrices due to thermal breakdown of cellular constituents [19].

Although the TGC of dried broccoli showed a decreasing trend for all drying treatments (Figure 2C), attributed to both enzymatic and non-enzymatic reactions during drying [2,40], the loss of glucosinolates was lesser at high drying temperatures (80 and 90 °C). It could be also inferred that fast heating in a short time promotes the preservation of TGC. According to the literature, high drying temperatures could inactivate the activity of the enzyme myrosinase and thereby retard the degradation of glucosinolates and retain an appreciable amount of them [19,41]. In such a case, if the product maintains a high glucosinolate level due to plant myrosinase enzyme inactivation, the conversion from glucosinolates to ITCs would take place in the colon where bacterial myrosinase hydrolyzes them into breakdown compounds and then are absorbed and/or excreted [14,42,43].

In addition, there are some works that study the influence of drying methods on the retention of glucosinolates. On the one hand, it has been shown that freeze-drying retains more TGC in broccoli than other drying methods, such as hot air drying, VD and microwave vacuum drying, due to the use of lower temperatures [18], while on the other hand, some studies have shown unexpected increments in certain glucosinolates from broccoli and cabbage [2,40], which probably might be related to the amino acids metabolism during drying, since amino acid are precursors for glucosinolate biosynthesis.

The antioxidant potential measured by 2,2-diphenyl-1-picryl-hydrazyl (DPPH) assay of the dried broccoli was also significantly reduced (*p* ˂ 0.05) compared to the fresh sample (Figure 2D). The DPPH value of fresh broccoli was 23.55 μmol trolox equivalents (TE)/g d.m. and varied between 10.23 and 13.35 μmol TE/g d.m. among the dried samples. A similar value was reported by Xu et al. [18] as 24.06 μmol TE/g d.m. for the fresh sample and also reported a decrease in the DPPH antioxidant capacity after subjecting the broccoli by hot air-, vacuum- and microwave-vacuum-drying. Based on the comparison between Figure 2A,B,D, it was observed that the changes in the DPPH•-scavenging activity of samples were followed by the changes in TPC and TFC. The correlation analysis showed that DPPH was positively correlated with TPC (0.747) and TFC (0.964). Apparently, flavonoids present in fresh and dried broccoli scavenge DPPH+ free radicals more effectively than other phenolic compounds.

On the other hand, the Oxygen Radical Absorbance Capacity (ORAC) value of fresh broccoli was 30.79 μmol TE/g d.m. and ranged between 73.38 and 84.58 μmol TE/g d.m. after drying. Our results are in agreement with those of Ninfali and Bacchiocca [44] who reported ORAC values of fresh broccoli as 3.35 μmol TE/g (on fresh weight). Other reported data for fresh broccoli provided 47.03 μmol TE/g d.m. [45] and 34.31 μmol TE/g d.m. [18]. Unlike DPPH, the antioxidant activity measured by the ORAC assay was significantly enhanced after drying (Figure 2E). The ORAC assay is an index for evaluating the degree of inhibition of peroxy-radical-induced oxidation of the antioxidant components present in the samples [46]. Based on the determination principle of ORAC, we speculated that, after drying, some phenolic components were partially oxidized, which can cause to have antioxidative features in comparison to unoxidized phenolic compounds of the fresh sample [39]. Other researchers suggest that this could be because the generation and accumulation of certain products due to the Maillard reaction during the drying process, enhancing the scavenging activity of free radicals [38,39]. In fact, there was a negative correlation between ORAC and TPC (−0.623) and TFC (−0.904), thereby corroborating that other compounds might be the main providers of the antioxidant activity in this assay.

### 2.4. Phenolic Profiles of Fresh and Powdered Broccoli

To further clarify changes in the phenolic compounds in broccoli, the identification and quantification of phenolics in the extracts were obtained by LC-MS/MS. Table 3 shows nine phenolic compounds identified but only five quantified in the dried broccoli. Chlorogenic acid, sinapic acid and caffeic acid were detected in all the studied extracts. Nevertheless, it did not detect ferulic acid in the extract obtained from the fresh sample and the concentration of coumaric acid was lower than the quantification limit in the extracts obtained from fresh and dried broccoli at 80 and 90 °C (Table 3). In the fresh samples, chlorogenic acid, sinapic acid and caffeic acid were the main phenolic compounds, with values of 53.61, 1.25 and 9.06 µg/g d.m., respectively. Similarly, Yılmaz et al. [35] and López-Hernández et al. [47] also reported that chlorogenic acid was the most dominant phenolic compound in fresh broccoli, showing similar values of 52.16 µg/g d.m. and 16.43 mg/kg (on fresh weight), respectively. The chlorogenic acid, sinapic acid and caffeic acid content increased significantly in comparison with the fresh one (*p* < 0.05) after drying. Especially for chlorogenic acid, its content was increased from 53.61 to 94.47 µg/g d.m. and, for sinapic acid, from 1.25 to 44.95 µg/g d.m. after VD. It is believed that the thermal action of drying may weaken the affinity between phenolics and cell structure (cell wall), thus increasing the availability for the extraction and quantification of free phenolic compounds and making their contents higher than those in the fresh samples [48]. It should also be noted that hydroxycinnamic acids, such as chlorogenic acid, sinapic acid and caffeic acid, have a stronger ability to bind hydrogen ions, being able to contribute to protection from oxidation of some flavonoids during drying [38]. Even though quercetin, kaempferol and isorhamnetin were not quantifiable in this study, some previous studies showed appreciable amounts of quercetin and kaempferol in broccoli [47,49]. These differences can be explained because the extracts used in the latter studies were subjected to hydrolysis treatment (bound forms) to hydrolyze the glycosylated flavonoids and form aglycone flavonols.

On the other hand, the correlation analysis showed that a negative correlation between DPPH and the main phenolic acids, i.e., chlorogenic acid (−0.687), sinapic acid (−0.734) and caffeic acid (−0.598), whereas ORAC was positively correlated with these compounds (0.781, 0.818 and 0.725, respectively). Some studies have reported that these phenolic acids apparently can undergo some derivation reactions caused by thermal action and generate some new phenolics, which may specifically bind to the intermediate Maillard reaction products [38,48]. Therefore, we speculate that these compounds might be the main contributors to the antioxidant activity in the ORAC assay.

### 2.5. Neuroprotective Effect of Fresh and Powdered Broccoli

Parkinson’s disease is the second most common neurodegenerative disorder characterized by the accumulation of Lewy bodies containing misfolded fibrillar α-synuclein (α-Syn), and its aggregation is a molecular mechanism involved in selective dopaminergic neurons’ death [9,50]. To test the neuroprotective effects of broccoli extracts, we carried out preliminary investigations to determine the effect of the mentioned extracts on the viability of SN4741 cells and to verify that these extracts do not exert a direct cytotoxic effect. First, we treated SN4741 cells with a range of 10, 50 and 100 µg/mL extract concentrations for 24 h and then evaluated the cell viability using the fluorescent probe SYTOX^®^ Green. Dimethyl sulfoxide (DMSO) was used as a negative control in a non-toxic concentration (Figure 3A). All extracts at higher concentrations (100 µg/mL) were found to have toxic effect after 24 h of treatment. Cells treated with the extracts obtained from fresh (10 µg/mL) and dried broccoli at 90 °C (50 µg/mL) also showed an increase in dead cells with respect to the control. However, all other treatments did not show an effect on cell viability compared with that of the control for those concentrations. A 50 µg/mL concentration of the extract was, therefore, used in subsequent experiments since this was the highest concentration that showed no cytotoxic effects on SN4741 cells (Figure 3A).

Broccoli extracts were used to evaluate neuroprotective effects in SN4741 cells expose to α-synuclein pre-formed fibrils (α-syn PFF) (1 µM/mL) (Figure 3B). As it was expected, α-syn PFF in the absence of extracts exhibited increase in cell death. In the presence of broccoli extracts, toxicity was reduced when compared to the α-syn PFF alone [50]. However, cells incubated with α-syn PFF in the presence of extracts obtained from dried broccoli at 50 and 60 °C demonstrated lower toxicity. It is noteworthy that the dried samples at 50 and 60 °C presenting low TGC values (Figure 2C), suggesting that enzymatic degradation of glucosinolates was produced, leading thereby to the formation of ITCs, implicated the activation of the nuclear factor (erythroid-derived 2)-like 2 (Nrf2), a master regulator of the antioxidant network and cytoprotective genes [9,10]. Therefore, based on the rationale that natural extracts that stimulate the Nrf2-mediated antioxidant pathway should have a high propensity to alleviate cell death triggered by oxidative insults [11], both extracts could induce a robust activation of Nrf2 signaling.

However, ITCs might not be the only candidates influencing the activation of Nrf2 signaling of broccoli extracts; other compounds, i.e., different classes of phenolic compounds, may also play such a role by acting synergistically with ITCs [51]. A positive correlation was actually found between the neuroprotective effect and TPC (0.776) as well as TFC (0.743). Nonetheless, a negative correlation with such effect and the main phenolic acids, i.e., chlorogenic acid (−0.440), sinapic acid (−0.191) and caffeic acid (−0.224), was found.

### 2.6. Antimicrobial Effect of Broccoli Powder

The available literature has indicated that broccoli’s crude extracts have antimicrobial effect against various pathogenic bacteria and phytophatogenic fungi [52], with hydrolytic products of glucosinolates, including thiocyanates, nitriles and ITCs, being responsible for these effects [53]. In the present study, we evaluated the susceptibility of *S. aureus*, *B. cereus*, *E. coli* and *S. typhimurium* against methanolic extracts obtained from vacuum-dried broccoli at different temperatures. It was shown that all bacteria were resistant to the extracts tested, as no inhibition halo was formed when performing the agar well diffusion method (data not shown). Due to the volatile nature of ITCs influenced by their structure (aliphatic and benzenic classes) [13], antimicrobial activity can be affected by the solvent extraction method [54]. Thus, to evaluate the antimicrobial effect of our samples, another extraction using pure supercritical CO_2_ fluid was conducted, and the zone inhibition diameters are summarized on Table 4. The antimicrobial effect of dried broccoli was evaluated by comparing different concentrations of extracts 50, 25, 12.5, 6.3, 3.1 and 1.6 mg/mL against all the tested cultures, where in turn we classified the resistance of the different microbial species to broccoli extracts based on the classification proposed by Valková et al. [55]. A halo size larger than 15 mm is considered a strong antimicrobial activity, while values between 11 and 14 mm had a moderate antimicrobial activity and between 5 and 10 mm showed a weak antimicrobial activity.

In general, *B.cereus* was the most sensitive of the tested bacteria toward the extracts studied, since, even at the lowest concentration (1.6 mg/mL), it produced zones of inhibition in some extracts (50 and 60 °C) (Table 4). This sensitivity might be because Gram-positive bacteria do not possess outer membrane protection. Therefore, the layer of peptidoglycans (mureins) and teichoic acids receives all the extracellular pressure and may easily be penetrated by antimicrobial agents from natural extracts [56,57]. In fact, Abukhabta et al. [58] reported that, from eleven studied strains, only *B. cereus* was inhibited by raw broccoli extracts. However, other Gram-positive bacteria as *S.aureus* may resist better the attack of natural antimicrobials [56]. These findings are consistent with the results of Pacheco-Cano et al. [52] for crude extracts of broccoli and Gudiño et al. [59] for broccoli stem ethanolic extracts, who also found higher inhibitory effect against *B. cereus* than against *S. aureus*.

The extracts (50 mg/mL) obtained from dried broccoli at 50, 60 and 70 °C have a strong activity against *S. typhimurium* and *S. aureus*, and a moderate activity against *E. coli*, whereas the extracts obtained from dried broccoli at 80 and 90 °C failed to produce inhibition zones against those strains. As already stated, a high drying temperature would have led to almost total myrosinase inactivation, which suggests that most of the glucosinolates present in the sample remained unhydrolyzed, providing a reason for the lack of an inhibitory effect of those extracts against those strains.

The highest antimicrobial effect was obtained in the extract from dried broccoli at 50 °C against all analyzed strains at the same concentration. It is noteworthy that the dried samples at 50 °C presented the lowest TGC (Figure 2C), suggesting that the hydrolysis of the glucosinolates, induced by the endogenous myrosinase activity that would have still been active in this sample, was produced, which could lead to the formation of ITCs implicated in the antimicrobial effect [13,53]. These results are corroborated by other authors who reported that an increase in antimicrobial activity by broccoli extracts could be interpreted due to the increased levels of ITCs, mainly sulforaphane, formed from glucoraphanin (~81% of total glucosinolate content in broccoli; [58]). However, in our study, there is a need to further explore the ITCs and other compounds present in broccoli extracts that may exert such antimicrobial effects.

## 3. Materials and Methods

### 3.1. Chemicals

All reagents used were of analytical or HPLC grade and were purchased from SIGMA-Aldrich (St. Louis, MO, USA): lithium chloride (LiCl), potassium acetate (CH_3_COOK), magnesium chloride (MgCl_2_), potassium carbonate (K_2_CO_3_), magnesium nitrate hexahydrate (Mg(NO_3_)_2_), sodium nitrate (NaNO_3_), potassium iodide (KI), sodium chloride (NaCl), potassium chloride (KCl), potassium nitrate (KNO_3_), potassium sulfate (K_2_SO_4_), methanol, Folin–Ciocalteu reagent, sodium tetrachloropalladate II (Na_2_PdCl_4_) reagent, 2,2-diphenyl-1-picrylhydrazyl (DPPH), gallic acid, quercetin, sinigrin, 6-Hydroxy-2,5,7,8-tetramethylchoman-2-carboxylic acid (Trolox), 2,2′-azobis (2-amidinopropane) dihydrochloride (AAPH), (0.1%) formic acid, (20%) sodium carbonate solution (Na_2_CO_3_), (10%) aluminum trichloride solution (AlCl_3_), (5%) sodium nitrite solution (NaNO_2_), sodium hydroxide (NaOH), potassium phosphate buffer, dimethyl sulfoxide (DMSO), Dulbecco’s modified Eagles medium (DMEM), fetal bovine serum (FBS), nutrient broth, Mueller–Hinton agar (MHA), Mueller–Hinton broth (MHB), tryptone soya broth (TSB), ferulic acid, chlorogenic acid, sinapic acid, caffeic acid, coumaric acid, cryptochlorogenic acid, kaempferol, isorhamnetin.

### 3.2. Raw Material and Sample Preparation

Fresh broccoli was provided by Dos Marías Company located in Pan de Azúcar, Coquimbo Region, Chile. Once in the laboratory, the broccoli was sorted visually for physical damage and kept in refrigerator at 4 °C for less than three days prior to further processing. The fresh broccoli heads were washed and drained. Then, the florets were cut into a uniform size without any bleaching pretreatment before drying. The initial moisture content of fresh broccoli was measured by AOAC (934.06) method. The value obtained was 87.25 ± 0.76% on a wet basis.

### 3.3. Vacuum-Drying Procedure

The drying of the broccoli was conducted by placing the small florets (250 g) in a thermoshelf of the vacuum dryer (Memmert, model VO 400, Schwabach, Germany). The oven is connected to a vacuum pump (Büchi, model V-100, Flawil, Switzerland), where vacuum inlet is controlled via solenoid valves. The pressure inside the system was set at 10 kPa. The broccoli florets were dried at five temperature levels of 50, 60, 70, 80 and 90 °C. The mass of the samples at a time t (M_t_) was recorded using a digital electronic balance (Ohaus SP402; precision ± 0.01 g) every 30 min throughout each drying experiment and stopped when there were no noticeable changes in mass of the samples for the last two data points (equilibrium moisture content). Experiments were conducted in triplicate.

### 3.4. Determination of the Desorption Isotherm

The desorption isotherms at 60 °C and 80 °C were obtained from fresh broccoli florets using the method recommended by Spiess and Wolf [60]. Saturated salt solutions of LiCl, CH_3_COOK, MgCl_2_, K_2_CO_3_, Mg(NO_3_)_2_, NaNO_3_, KI, NaCl, KCl, KNO_3_ and K_2_SO_4_ were prepared and placed in different hermetic glass recipient. Three broccoli florets were placed in Petri dishes and carried into each recipient. Meanwhile, a small amount of thymol was placed in recipients with $a_{w}$ higher than 0.60 for inhibit the microbial growth on broccoli samples. Sample weight was measured weekly using an analytical balance (±0.0001 g; HR200, A&D Company, Tokyo, Japan) until the equilibrium of the mass was reached. Then, water contents of equilibrated samples were measured by the AOAC (934.06) method in triplicate.

The experimental data were then fitted to BET model (Equation (1)), so as to build a quantitative relationship between $a_{w}$ and equilibrium moisture content [40]:(2)$$X_{we}=\frac{a_{w}\cdot X_{m}\cdot C}{\left(1-a_{w}\right)\cdot \left(a_{w}\cdot \left(C-1\right)+1\right)}$$
where $X_{we}$ (g water/g dry matter (d.m.)) is the equilibrium moisture content; $X_{m}$ (g water/g d.m.) represents the moisture content of the monolayer; C is the energetic constant model; and $a_{w}$ is the water activity (dimensionless).

### 3.5. Drying Kinetics Modeling

The drying curves were drawn from the experimental results and the moisture ratio (MR) was determined using Equation (2).
(3)$$MR=\frac{X_{wt}-X_{we}}{X_{w0}-X_{we}}$$
where $X_{wt}$ (g water/g d.m.) is the moisture at time t; $X_{we}$ (g water/g d.m.) is the equilibrium moisture content; and $X_{w0}$ (g water/g d.m.) is the initial moisture. Seven mathematical models were used to fit the experimental data of moisture loss during the drying process through predicting the *MR* at each drying temperature. The equation models were obtained from Inyang et al. [61] and are presented in Table 1.

### 3.6. Moisture Content, $a_{w}$ and Proximate Composition

Moisture, fat, ash, protein and crude fiber contents were determined using standard methods [62]. $a_{w}$ was determined with a water activity analysis instrument (AquaLab 4 TE, Pullman, WA, USA) at 25 °C. All measurements were performed in triplicate.

### 3.7. Determination of Bioactive Compounds and Antioxidant Potential

#### 3.7.1. Preparation of Broccoli Extracts

The extraction process of the bioactive compounds of broccoli was conducted by means of the method proposed by Ke et al. [63]. Fresh and dried broccoli samples were thoroughly dissolved in 80% aqueous methanol at a 1:5 and 1:10 *w*/*v* ratio, respectively, and agitated in an orbital shaker (OS-20, Boeco, Hamburg, Germany) at 250 rpm for 1 h, followed by centrifugation at 4193× *g* for 20 min at 4 °C (5804 R, Eppendorf, Hamburg, Germany). The supernatant was filtered, and the solid residue was re-extracted twice. The combined filtered extracts combined were evaporated by a multivapor (Büchi, P-6, Flawil, Switzerland) at 40 °C and then lyophilized. The final lyophilized residue was reconstituted in 5 mL of a methanol–formic acid solution (99:1) to carry out further analyses.

#### 3.7.2. Determination of Total Phenolic and Flavonoid Contents

The total phenolic content (TPC) of extracts was estimated as gallic acid equivalents (GAE) using the Folin–Ciocalteu method, whereas the total flavonoids content (TFC) was measured according to a colorimetric assay and estimated as quercetin equivalents (QE). The absorbance of the extracts was determined at 725 nm for TPC and 415 nm for TFC using a Vis spectrophotometer (Spectronic Instruments, Spectronic 20 Genesys, Rochester, NY, USA). TPC and TFC were expressed as mg GAEor QE per g of sample (d.m.), respectively. The detailed methods have been described by Puente et al. [64].

#### 3.7.3. Determination of Total Glucosinolate Content

Total glucosinolate content (TGC) was estimated using the sodium tetrachloropalladate II (Na_2_PdCl_4_) method [65]. Briefly, an aliquot of 60 μL of each extract solution was transferred to a test tube with 1800 μL of Na_2_PdCl_4_ reagent (2 mM) and incubated in the dark at room temperature for 30 min. After that, the absorbance at 450 nm was read. The TGC was based on the standard curve of Sinigrin (Sigma–Aldrich, S1647, St. Louis, MO, USA) and results were expressed as μmol SE per g of sample (d.m.).

#### 3.7.4. Antioxidant Potential Measurement

The DPPH radical scavenging activity and ORAC of the broccoli samples were measured according to the previously detailed method described by Puente et al. [64]. The results were expressed as μmol TE/g d.m.

#### 3.7.5. Chromatographic Analysis

Individual phenolic components were qualitatively determined using an ABSciex Triple Quad™ 4500 LC-MS/MS System (AB/Sciex, Concord, ON, Canada) operated with an electrospray (TurboV) source. The separation was performed using the Eksigent Ekspert Ultra LC100 system with an LiChrospher 100 RP-18 column (125 mm × 4 mm i.d., 5 µm; Merck, Darmstadt, Germany) and an Ekspert Ultra LC100-XL autosampler. Both electrospray in the negative mode and chromatographic separation were carried out employing the conditions described by de Camargo et al. [66]. The column temperature was maintained at 50 °C, the injection volume was 10 μL and a flow rate of 0.5 mL/min. All compounds were identified by comparing dynamic multiple reaction monitoring information, including retention time, MS and fragment ions. Calibration curves for quantification were built utilizing commercially available standards: Ferulic acid (y = 15,634x + 3 × 10^6^); Chlorogenic acid (y = 24,456x + 2 × 10^6^); Sinapic acid (y = 8718.1x + 525,181); Caffeic acid (y = 45,273x + 4 × 10^6^); Coumaric acid (y = 41,289x + 1 × 10^7^); Quercetin (y = 4012.6x + 522,991); Kaempferol (y = 1194.6x + 351,002); and Isorhamnetin (y = 8112.9x + 3 × 10^6^).

### 3.8. Neuroprotective Potential

#### 3.8.1. Cell Culture

Mouse dopaminergic neuron SN4741 cell lines were purchased from ATCC (Manassas, VA, USA). Cell lines were cultured in DMEM with 1% penicillin/streptomycin and 10% FBS (Life Technologies, Thermo Fisher Scientific, Waltham, MA, USA) and maintained at 37 °C and 5% CO_2_.

#### 3.8.2. Cell Viability Assay

The cells were seeded in a 96-well plate at a confluence of 1 × 10^4^ cells per well in 100 µL of medium (DMEM + 10% FBS). Once the cells were adhered, they were washed and treated with α-syn PFF (1 µM/mL). The broccoli extracts were added together with α-syn PFF at 50 µM. In addition, SYTOX^®^ Green 5 µM was added to determine dead cell viability. After 24 h, cell viability was analyzed by fluorescence (Excitation/Emission (nm) 504/523).

### 3.9. Extraction and Antimicrobial Potential

#### 3.9.1. Preparation of the Broccoli Extracts by Supercritical Fluid Extraction (SFE)

Extractions were carried out as according to the method of Goyeheche et al. [67] with modifications. Five grams of vacuum dried broccoli were separated aside for each experiment. Each extraction was carried out using pure supercritical CO_2_ fluid in a Spe-ed SFE equipment (Model 7070, Allentown, PA, USA). Briefly, samples were mixed with 6 mL of 5% ethanol (used as a co-solvent) and then were loaded into a 50 mL steel cylindrical extractor vessel. After the loaded extractor vessel was assembled, CO_2_ was pumped into the extractor vessel with an ISCO 500D syringe pump. Static extraction assays were performed at 25 MPa and 35 °C with a CO_2_ flow rate of 1 mL/min at constant pressure and an extraction time of 30 min. The dynamic extraction time was 2 h and 30 min. When the programmed time was reached, the extractor vessel was depressurized and the extract was separated from the carbon dioxide and collected in a glass tube at 100 °C to prevent the CO_2_ from freezing. The recently obtained extracts were immediately stored in a freezer at −20 °C until use. All extractions were carried out in triplicate.

#### 3.9.2. Culture Maintenance and Inoculum Preparation

The strains *Staphylococcus aureus* (ATCC 25923), *Bacillus cereus* (ATCC 10876), *Escherichia coli* (ATCC 25922) and *Salmonella typhimurium* (ATCC 13311) were used in this step. For each strain, a 20% glycerol stock culture was maintained on a nutrient broth (Difco) at −80 °C. Before they were used, the cultures were transferred to Petri plates with MHA (Oxoid) and incubated at 37 °C. From these plates, 1–2 isolated colonies were cultured to 5 mL of MHB (Merck) for 48 h at 37 °C. Cultures were then transferred to TSB (Difco) incubated for 12 to 18 h and used as inoculum source for each experiment. Inoculums were standardized at a McFarland turbidity of 0.5 (corresponding to 10^7^ CFU/mL) for bacteria.

#### 3.9.3. Antimicrobial Zone Inhibition Assay

Initially, the broccoli extracts were solubilized in pure DMSO and their antimicrobial potential was tested using the agar well-diffusion method recommended by the Clinical and Laboratory Standards Institute [68]. Briefly, 100 µL of each standardized inoculum of the respective microbial strain was plated on the surface of MHA. Then, 10 μL of each broccoli extract, tested at the final concentrations of 50, 25, 12.5, 6.3, 3.1 and 1.6 mg/mL, were pipetted separately into sterilized filter paper discs (6 mm in diameter) and placed on the agar and the plates were incubated at 37 °C for 24 h. Amoxicillin + clavulanic acid (100 µg/mL) was used as positive control for *S. aureus* and ciprofloxacin (100 µg/mL) for the other tested bacteria, while pure DMSO was used as the negative control. The zones of inhibition (transparent zones around the wells) were measured for their diameter (mm).

### 3.10. Statistical Analysis

The results were expressed as means ± standard deviation (SD) and statistically analyzed using the Origin software Version 8 (OriginLab Corporation, Northampton, MA, USA). The means were compared by an analysis of variance (ANOVA) and Tukey’s test to estimate the significance among the main effects at the 5% probability level. Pearson’s correlation coefficients were calculated using Microsoft Office Excel 2013 function. The model parameters determination was carried out through an iterative method implemented in the statistic free software Rstudio by Dose-Response Analysis (drc) library and Nonlinear Least Squares with Brute Force (nls2) library.

The quality fit of the BET model and kinetic mathematical models were evaluated by the determination coefficient (*R*^2^), sum of squared errors (*SSE*) and chi-squared (*χ*^2^), described below:(4)$$R^{2}=1-\frac{\sum_{i=1}^{N}{\left(Exp_{i}-Cal_{i}\right)}^{2}}{\sum_{i=1}^{N}{\left(Exp_{i}-\bar{Exp_{i}}\right)}^{2}}$$
(5)$$SSE=\frac{1}{N}\overset{N}{\underset{i=1}{\sum }}{\left(Exp_{i}-Cal_{i}\right)}^{2}$$
(6)$$\chi^{2}=\frac{\sum_{i=1}^{N}{\left(Exp_{i}-Cal_{i}\right)}^{2}}{N-z}$$
where $Exp_{i}$ is the experimental data, $Cal_{i}$ is the calculated values, and z is the model constant numbers.

## 4. Conclusions

In this study, we presented VD as an alternative method for use in the industrial food manufacturing. The MR of broccoli during the VD process was found to be exponentially decreasing with increase in drying times. Additionally, it was determined that the drying kinetic parameters were best defined by the model of Midilli and Kucuk. The applications of higher VD temperatures (80 and 90 °C) are good to retain the TPC and TFC in the dried samples, both contents being positively correlated with the antioxidant capacity by the DPPH assay. Preliminarily, this study revealed that hydroxycinnamic acids, such as chlorogenic acid, sinapic acid and caffeic acid, are correlated with an increase in antioxidant activity by the ORAC assay after the drying of the broccoli. This suggests that these acids might be involved in the Maillard reaction caused by thermal action. Furthermore, some lower VD temperatures (especially at 50 °C) were found to provide a reduced amount of the TGC, suggesting that the enzymatic degradation of glucosinolates was produced with a simultaneously formation of ITCs, whose compounds are involved in the neuroprotective and antimicrobial effects of broccoli. In this study, the exact mechanism of the neuroprotective and antimicrobial behavior was not fully understood, and further research is required. However, the results of this study are important to clarify the effect of dried broccoli consumption on Parkinson’s disease, as they do indicate that cells incubated with α-syn PFF in the presence of broccoli extracts have a lower toxicity, suggesting the potential of broccoli as adjuvants for the treatment of Parkinson’s disease. However, they must be verified considering the eventual interactions occurring among broccoli components, especially ITCs. Therefore, a goal for future studies is to identify and quantify the ITCs from vacuum-dried broccoli to take advantage of their therapeutic effects.

## Figures and Tables

**Figure 1 molecules-28-00766-f001:**
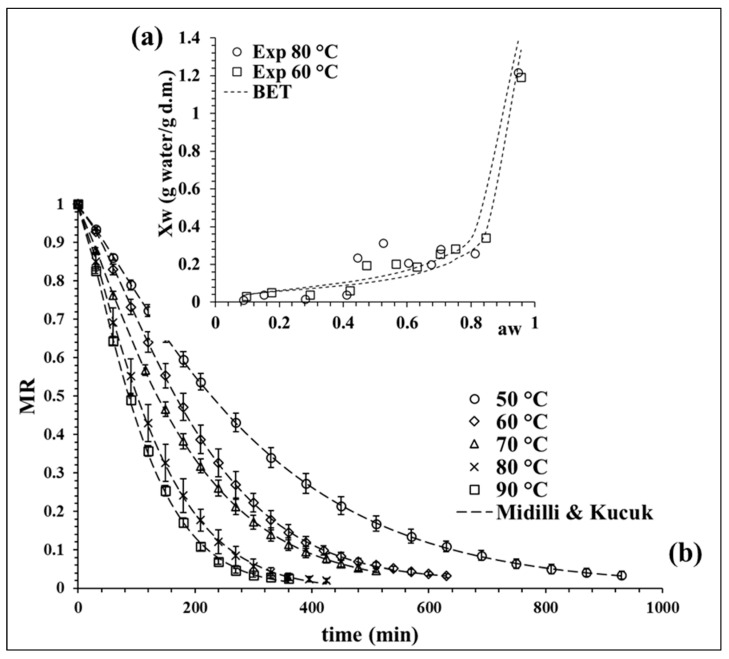
(**a**) BET model isotherm at 60 and 80 °C, and (**b**) drying kinetic of broccoli at different temperatures with the fit of Midilli and Kucuk model. Values are averages (*n* = 3), and error bars are standard deviation.

**Figure 2 molecules-28-00766-f002:**
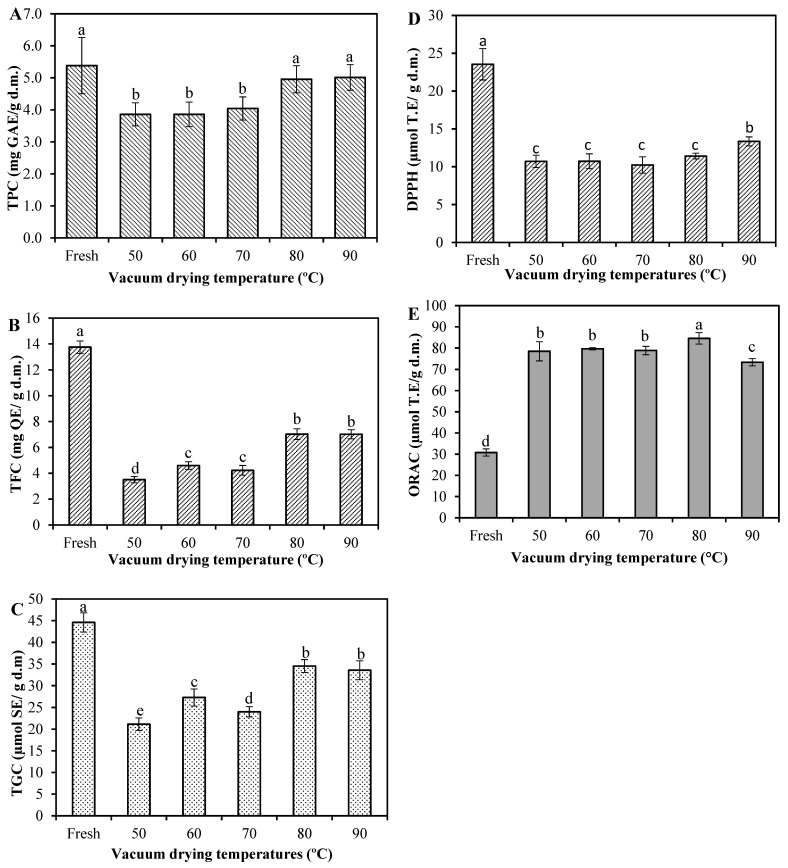
Effects of vacuum-drying temperature on (**A**) total phenolic content (TPC); (**B**) total flavonoid content (TFC); (**C**) total glucocinolate content (TGC); (**D**) 2,2-diphenyl-1-picryl-hydrazyl (DPPH) and (**E**) Oxygen Radical Absorbance Capacity (ORAC) in broccoli. Values are means of triplicate analyses (*n* = 3) and error bars are standard deviation. On the bars, different letters (a, b, c, d and e) indicate significant differences as per Tukey’s Multiple Range Test (*p* < 0.05). Gallic acid equivalents (GAE); Quercetin equivalents (QE); Sinigrin equivalents (SE); and Trolox equivalents (TE).

**Figure 3 molecules-28-00766-f003:**
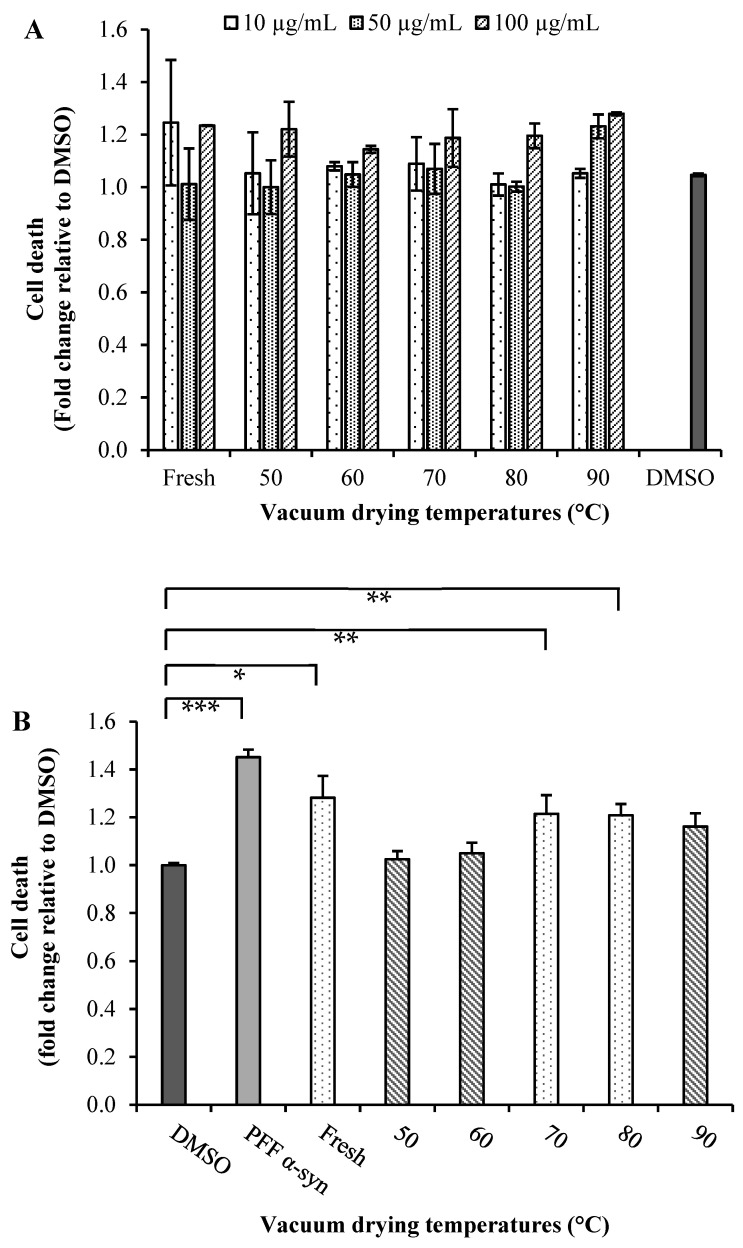
(**A**) Cytotoxicity and (**B**) neuroprotective effects of extracts of broccoli: SN741 cells were treated with of α-synuclein pre-formed fibrils (α-syn PFF) and broccoli extract obtained at different temperatures (50, 60, 70, 80 and 90 °C) by 24 h. DMSO was used as the control. After treatment, we determined the cell viability using SYTOX^®^ Green probe. Data are presented as mean and SEM of three independent experiments performed in triplicate. Statistically significant differences were detected by an ordinary one-way ANOVA (***: *p* < 0.001; **: *p* < 0.01; *: *p* < 0.05).

**Table 1 molecules-28-00766-t001:** Coefficients of the mathematical models applied to the drying curves of broccoli.

Model	Parameters	Vacuum-Drying Temperatures (°C)				
		50	60	70	80	90
Midilli–Kucuk	Equation	MR = a exp(−kt^n^) + bt				
	a	0.997300 ± 0.000100	0.995000 ± 0.002858	0.995367 ± 0.001922	0.995837 ± 0.003986	0.994267 ± 0.000153
	k	0.001129 ± 0.000092	0.000695 ± 0.000140	0.002098 ± 0.000057	0.001982 ± 0.000655	0.002154 ± 0.000138
	n	1.182333 ± 0.007371	1.353667 ± 0.021548	1.180333 ± 0.014295	1.274667 ± 0.000012	1.293000 ± 0.010440
	b	0.000008 ± 0.000006	0.000029 ± 0.000004	0.000011 ± 0.000005	−0.000005 ± 0.000012	0.000012 ± 0.000014
	R^2^	0.99994	0.99983	0.99986	0.99965	0.99951
	SSE	0.0000057	0.0000120	0.0000107	0.0000291	0.0000477
	χ^2^	0.000007	0.000015	0.000014	0.000040	0.000069
Logarithmic	Equation	MR = a exp(−kt) + c				
	a	1.084382 ± 0.006208	1.136048 ± 0.021279	1.078490 ± 0.003831	1.111592 ± 0.033180	1.104668 ± 0.012193
	k	0.002983 ± 0.000199	0.004210 ± 0.000451	0.004997 ± 0.000287	0.006799 ± 0.001016	0.007945 ± 0.000368
	c	−0.052828 ± 0.007588	−0.079400 ± 0.022543	−0.055687 ± 0.004800	−0.076427 ± 0.029019	−0.071284 ± 0.011698
	R^2^	0.99807	0.99086	0.99764	0.98644	0.99448
	SSE	0.0001848	0.0005740	0.0001487	0.0004248	0.0005289
	χ^2^	0.000217	0.000665	0.000181	0.000531	0.000688
Silva & Alii	Equation	MR = exp(−at − b√t)				
	a	0.004029 ± 0.000236	0.006686 ± 0.000403	0.006787 ± 0.000369	0.010668 ± 0.000871	0.012610 ± 0.000396
	b	−0.013837 ± 0.000547	−0.029878 ± 0.001048	−0.017970 ± 0.000995	−0.034889 ± 0.004209	−0.039513 ± 0.002219
	R^2^	0.99931	0.99791	0.99914	0.99766	0.99786
	SSE	0.0000694	0.0001857	0.0000728	0.0002105	0.0002111
	χ^2^	0.000077	0.000204	0.000082	0.000243	0.000249
Weibull	Equation	MR = exp [−(t/β)^α^]				
	β	313.7490 ± 18.25255	221.2163 ± 14.40987	186.4917 ± 8.866578	135.2960 ± 13.33205	115.4380 ± 3.220747
	α	1.164667 ± 0.003055	1.299667 ± 0.023072	1.160000 ± 0.006000	1.268667 ± 0.055752	1.272333 ± 0.020033
	R^2^	0.99989	0.99949	0.99983	0.99959	0.99945
	SSE	0.0000088	0.0000410	0.0000132	0.0000317	0.0000515
	χ^2^	0.000010	0.000045	0.000015	0.000037	0.000061
Newton	Equation	MR = exp(−kt)				
	k	0.003202 ± 0.000183	0.004686 ± 0.000317	0.005525 ± 0.000277	0.007725 ± 0.000800	0.009007 ± 0.000254
	R^2^	0.99307	0.98116	0.99397	0.98572	0.98605
	SSE	0.0007233	0.0017654	0.0005253	0.0013586	0.0014010
	χ^2^	0.000761	0.001849	0.000558	0.001456	0.001518
Page	Equation	MR = exp(−kt^n^)				
	k	0.001239 ± 0.000080	0.000909 ± 0.000178	0.002327 ± 0.000056	0.002072 ± 0.000732	0.002384 ± 0.000257
	n	1.164833 ± 0.003280	1.299710 ± 0.023105	1.159854 ± 0.005859	1.268532 ± 0.055616	1.272576 ± 0.019979
	R^2^	0.99989	0.99949	0.99983	0.99959	0.99945
	SSE	0.0000079	0.0000318	0.0000132	0.0000317	0.0000515
	χ^2^	0.000009	0.000035	0.000015	0.000037	0.000061
Modified Page	Equation	MR = exp[−(kt)^n^]				
	k	0.003194 ± 0.000179	0.004534 ± 0.000303	0.005370 ± 0.000253	0.007442 ± 0.000777	0.008667 ± 0.000245
	n	1.164834 ± 0.003279	1.299710 ± 0.023106	1.159854 ± 0.005859	1.268530 ± 0.055616	1.272576 ± 0.019978
	R^2^	0.99989	0.99949	0.99983	0.99959	0.99945
	SSE	0.0000088	0.0000411	0.0000132	0.0000317	0.0000515
	χ^2^	0.000010	0.000045	0.000015	0.000037	0.000061

**Table 2 molecules-28-00766-t002:** Effects of the vacuum-drying temperature on moisture content, water activity and proximate composition of broccoli.

Parameters	Vacuum-Drying Temperature (°C)					
	Fresh	50	60	70	80	90
^1^ Moisture	87.25 ± 0.76 ^a^	10.09 ± 0.12 ^b^	9.69 ± 0.21 ^b^	7.03 ± 0.04 ^c^	6.23 ± 0.17 ^d^	4.95 ± 0.18 ^e^
^2^ Water activity ($a_{w}$)	0.9893 ± 0.0001 ^a^	0.4695 ± 0.0028 ^b^	0.4394 ± 0.0011 ^c^	0.2915 ± 0.0016 ^d^	0.2442 ± 0.0022 ^e^	0.2131 ± 0.0049 ^f^
^3^ Fat	1.18 ± 0.02 ^c^	4.13 ± 0.07 ^b^	4.69 ± 0.24 ^a^	4.63 ± 0.41 ^a^	4.37 ± 0.07 ^ab^	4.36 ± 0.13 ^ab^
^3^ Ash	11.25 ± 1.16 ^a^	9.30 ± 0.03 ^b^	8.81 ± 0.38 ^b^	8.76 ± 0.06 ^b^	8.35 ± 0.37 ^b^	8.72 ± 0.06 ^b^
^3^ Crude protein	29.91 ± 1.94 ^d^	36.00 ± 0.26 ^a^	33.26 ± 0.33 ^b^	36.54 ± 0.17 ^a^	34.19 ± 0.27 ^b^	31.46 ± 0.15 ^c^
^3^ Crude fiber	11.47 ± 0.25 ^a^	9.97 ± 0.63 ^c^	8.67 ± 0.67 ^c^	8.86 ± 0.60 ^c^	9.21 ± 0.20 ^bc^	9.13 ± 0.44 ^bc^

Values are expressed as mean ± standard deviation of triplicate measurements (*n* = 3). In a row, different letters (^a, b, c, d, e^ and ^f^) indicate significant differences as per Tukey’s Multiple Range Test (*p* < 0.05). ^1^ Expressed as g/100 g. ^2^ Dimensionless. ^3^ Expressed as g/100 g d.m.

**Table 3 molecules-28-00766-t003:** Effect of vacuum-drying temperature on phenolic compounds profile in broccoli (µg/g d.m.) determined by LC-MS/MS.

Parameters	Vacuum-Drying Temperature (°C)					
	Fresh	50	60	70	80	90
Ferulic acid	ND	3.45 ± 0.17 ^b^	1.15 ± 0.21 ^e^	2.70 ± 0.09 ^c^	3.96 ± 0.32 ^a^	1.89 ± 0.20 ^d^
Chlorogenic acid	53.61 ± 3.64 ^d^	88.79 ± 5.21 ^ab^	72.01 ± 3.89 ^c^	70.31 ± 2.01 ^c^	94.47 ± 4.89 ^a^	87.80 ± 0.50 ^b^
Sinapic acid	1.25 ± 0.13 ^e^	28.38 ± 3.05 ^b^	16.22 ± 2.33 ^d^	26.31 ± 0.19 ^b^	44.95 ± 0.90 ^a^	22.91 ± 0.26 ^c^
Caffeic acid	9.06 ± 0.36 ^d^	11.71 ± 1.22 ^cd^	14.71 ± 2.00 ^ab^	12.04 ± 2.00 ^bc^	17.03 ± 2.25 ^a^	15.98 ± 0.50 ^a^
Coumaric acid	LLOQ	3.58 ± 0.75 ^a^	0.53 ± 0.10 ^b^	0.85 ± 0.21 ^b^	LLOQ	LLOQ
Cryptochlorogenic acid	LLOQ	LLOQ	LLOQ	LLOQ	LLOQ	LLOQ
Quercetin	LLOQ	LLOQ	LLOQ	LLOQ	LLOQ	LLOQ
Kaempferol	ND	LLOQ	LLOQ	LLOQ	LLOQ	LLOQ
Isorhamnetin	LLOQ	LLOQ	LLOQ	LLOQ	LLOQ	LLOQ

Values are expressed as mean ± standard deviation of triplicate measurements (*n* = 3). In a row, different letters (^a, b, c, d^ and ^e^) indicate significant differences as per Tukey’s Multiple Range Test (*p* < 0.05). ND: not detected; LLOQ: lower limit of quantification.

**Table 4 molecules-28-00766-t004:** Inhibition zone (in mm) of the disk diffusion method that evaluates the antimicrobial activity of vacuum-dried broccoli extracts at five different temperatures and the effect of antibiotics against four bacterial strains.

Bacterial Strain	Concentration	Vacuum-Drying Temperature (°C)					^3^ Negative	* Positive
	mg/mL	50	60	70	80	90	Control	Control
^1^ *Salmonella typhimurium*	50.0	18.6 ± 0.6 ^b^	15.7 ± 1.8 ^c^	17.5 ± 0.9 ^bc^	0.00 ± 0.0 ^d^	0.00 ± 0.0 ^d^	0.00 ± 0.0 ^d^	49.8 ± 0.2 ^a^
	25.0	12.5 ± 0.5 ^b^	0.00 ± 0.0 ^c^	13.0 ± 1.0 ^b^	0.00 ± 0.0 ^c^	0.00 ± 0.0 ^c^	0.00 ± 0.0 ^c^	49.8 ± 0.2 ^a^
^1^ *Escherichia coli*	50.0	13.4 ± 0.4 ^b^	12.9 ± 1.2 ^b^	11.6 ± 0.5 ^c^	0.00 ± 0.0 ^d^	0.00 ± 0.0 ^d^	0.00 ± 0.0 ^d^	34.3 ± 0.2 ^a^
	25.0	10.7 ± 0.5 ^b^	9.3 ± 0.4 ^c^	8.8 ± 0.2 ^c^	0.00 ± 0.0 ^d^	0.00 ± 0.0 ^d^	0.00 ± 0.0 ^d^	34.3 ± 0.2 ^a^
^2^ *Staphylococcus aureus*	50.0	16.5 ± 0.5 ^b^	15.4 ± 0.4 ^c^	15.7 ± 0.6 ^bc^	0.00 ± 0.0 ^d^	0.00 ± 0.0 ^d^	0.00 ± 0.0 ^d^	33.4 ± 0.3 ^a^
	25.0	11.9 ± 0.1 ^b^	0.00 ± 0.0 ^d^	10.5 ± 0.5 ^c^	0.00 ± 0.0 ^d^	0.00 ± 0.0 ^d^	0.00 ± 0.0 ^d^	33.4 ± 0.3 ^a^
	12.5	8.8 ± 0.3 ^b^	0.00 ± 0.0 ^c^	0.00 ± 0.0 ^c^	0.00 ± 0.0 ^c^	0.00 ± 0.0 ^c^	0.00 ± 0.0 ^c^	33.4 ± 0.3 ^a^
^2^ *Bacillus cereus*	50.0	19.6 ± 0.5 ^b^	14.3 ± 0.7 ^c^	14.8 ± 0.3 ^c^	11.4 ± 1.0 ^d^	9.0 ± 0.0 ^e^	0.00 ± 0.0 ^f^	33.6 ± 0.5 ^a^
	25.0	13.0 ± 0.5 ^b^	12.1 ± 0.6 ^bc^	11.3 ± 0.6 ^c^	10.0 ± 1.0 ^d^	8.5 ± 0.5 ^e^	0.00 ± 0.0 ^f^	33.6 ± 0.5 ^a^
	12.5	9.5 ± 0.9 ^b^	9.3 ± 0.8 ^bc^	8.3 ± 0.5 ^c^	8.9 ± 0.2 ^bc^	8.1 ± 1.0 ^c^	0.00 ± 0.0 ^d^	33.6 ± 0.5 ^a^
	6.3	8.5 ± 0.1 ^b^	8.2 ± 0.3 ^bc^	7.6 ± 0.4 ^c^	7.7 ± 0.6 ^c^	7.6 ± 0.7 ^c^	0.00 ± 0.0 ^d^	33.6 ± 0.5 ^a^
	3.1	7.5 ± 0.1 ^b^	7.3 ± 0.6 ^b^	7.0 ± 0.0 ^b^	7.2 ± 0.3 ^b^	0.00 ± 0.0 ^c^	0.00 ± 0.0 ^c^	33.6 ± 0.5 ^a^
	1.6	7.3 ± 0.5 ^b^	7.0 ± 0.0 ^b^	0.00 ± 0.0 ^c^	0.00 ± 0.0 ^c^	0.00 ± 0.0 ^c^	0.00 ± 0.0 ^c^	33.6 ± 0.5 ^a^

Values are expressed as mean ± standard deviation of triplicate measurements (*n* = 3). In a row, different letters (^a, b, c, d, e^ and ^f^) indicate significant differences as per Tukey’s Multiple Range Test (*p* < 0.05). ^1^ Gram-negative bacteria. ^2^ Gram-positive bacteria. ^3^ Negative control: pure DMSO. * Ciprofloxacin (100 µg/mL) was used as positive control for *S. typhimurium*, *E. coli* and B. *cereus*, while Amoxicillin + clavulanic acid (100 µg/mL) was used for *S. aureus*.

## Data Availability

The data are contained within the article.

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
