# Peer review of "Comprehensive Evaluation of the Bioactive Composition and Neuroprotective and Antimicrobial Properties of Vacuum-Dried Broccoli (Brassica oleracea var. italica) Powder and Its Antioxidants"

_molecules, 2023, doi:10.3390/molecules28020766_

Round 1

Reviewer 1 Report

The manuscript was good writing and presentation. However, some points of the manuscript need to clarify and revise via the attachment comment as a PDF file. Please check it out.

Author Response

Reviewer#1

The manuscript was good writing and presentation. However, some points of the manuscript need to clarify and revise via the attachment comment as a PDF file. Please check it out.

The topic “Comprehensive evaluation of bioactive composition and its antioxidant, neuroprotective and antimicrobial properties of vacuum-dried broccoli (Brassica oleracea var. italica) powder” is an interesting and novelty moderate of study in the field. However, before accepting for publication, some point of manuscript needs to revise and make it clear for correct understanding.

Authors: We are grateful for the helpful comments that reviewer #1 has made. Please find below our itemized responses to your comments.

  1. Typographical errors exist throughout the manuscript. Rectify carefully.

Authors: We agree, it has been rectified typographic errors in whole manuscript according to the reviewer’s recommendation.

  1. What is the objective for including the BET model isotherm in the manuscript? Why the author has studied the term of sorption isotherm prediction with mathematic model for powder sample?

Authors: As we have explained in the manuscript, it was determined desorption isotherms at 60 °C and 80 °C using fresh broccoli florets (not powder samples). In drying operation the removal of water is important, and hence the desorption equilibrium moisture relationship is required to determine the lowest attainable moisture content and relative humidity at the same process temperature.

On the other hand, the precise determination of equilibrium moisture contents of dehydrated foods provides valuable information for the accurate computation of moisture ratio (MR), indicating the effectiveness of a theoretical, semi-theoretical or empirical model to fit the equilibrium moisture content data of agricultural products. The BET isotherm equation is one of the most widely used models and gives good fit for a variety of foods over the region 0.05 < aw < 0.45, obtaining in our study average values of 0.87; 0.0255 and 0.0081 for R2, SSE and χ2, respectively.

  1. What is the reason behind the choice of selected bacterial cultures concentration for the study? It seems only bacillus cereus was not similar with other?

Authors: As the concentration of antimicrobial compounds in vegetables extracts is likely low, it is noteworthy to mention that the observed antimicrobial effects are not comparable with those of pure antibiotics or antimicrobials. Therefore, we classified the resistance of the different microbial strains to broccoli extracts based on the classification proposed by Valková et al. (2022). A halo size larger than 15 mm is considered a strong antimicrobial activity; while values between 11 and 14 mm is considered a moderate antimicrobial activity and between 5 and 10 mm a weak antimicrobial activity. Based on this classification, preliminary tests in our laboratory and literature, we selected 50 mg/mL as the maximum concentration. Also, the addition of natural preservatives such as plant extracts often rises the cost of a food product, since they need to be subjected to extraction and purification procedures that make natural preservatives more expensive than the artificial ones (Olszewska, et al., 2020). For this reason, we also tested lower concentrations of extracts (25, 12.5, 6.3, 3.1 and 1.6 mg/mL). However, only B. cereus was sensitive even at the lowest concentration (1.6 mg/mL) produced zones of inhibition in some extracts (50 and 60 °C) (Table 4). This sensitivity might be because the Gram-positive bacteria do not possess outer membrane protection. Therefore, the layer of peptidoglycans (mureins) and teichoic acids receives all the extracellular pressure and may easily be penetrated by antimicrobial agents from natural extracts (Maria-Neto et al., 2015; Mongalo et al., 2022).

  1. In insecticidal activity, whether the test insect was of similar stages (e.g. 3rd instar, pupa, adult) or was it a heterogeneous colony? It is unclear and needs clarification

Authors: Sorry, but there must be a confusion. The "insecticidal activity" was not measured in this study.

  1. The research article addresses on the application of vacuum-dried broccoli (Brassica oleracea var. italica) powder or its biological activity on the neuro system if they had to enter the blood systems what will be the ways of elimination or fate of disposal from the biological food chain. Justify.

Authors: Thank you very much for the reviewer's comments. The broccoli extract or its bioactive components could cross the intestinal barrier and being delivered to the bloodstream and subsequently cross the blood-brain barrier and exert its local effect in the brain tissue. Regarding its biological elimination, it will probably be through the fecal route like other foods of plant origin. Nonetheless, it should be verified in future studies.

  1. Check the abbreviations throughout the manuscript and introduce the abbreviation when the full word appears the first time in the abstract and the remaining for the text and then use only the abbreviation. Make a word abbreviated in the article that is repeated at least three times in the text, not all words need to be abbreviated.

Authors: We have checked the abbreviations in the whole manuscript and corrected them, according to the reviewer’s recommendation. However, in the figures y tables captions, the full word has been maintained and the abbreviations as well as in some materials and methods sections for ease of reading. We hope you agree.

  1. The authors should include the source of chemicals used in the present investigation in the materials and methods section.

Authors: We agree, it has been included a subchapter of the chemicals used in the present investigation in the materials and methods section according to the reviewer’s recommendation.

  1. In the conclusion seems to be in general and is not given separately, it is highly recommended to include limitation of the study and potential future research goals.

Authors: We agree, it has been included some limitations of this study and potential future research goals according to the reviewer’s recommendation.

Reviewer 2 Report

The information in the introduction is really scattered. "Despite broccoli popularity..." There is no consumer data used by the Authors to confirm that statement. Line 53: the idea of broccoli powders as a functional food doesn't fit the info of broccoli cultivation and therefore, with the idea of making powders. The aim of this research is not well presented. Furthermore, drying is an expensive method and uses a lot of energy. According to the energy crisis is there any issue that vacuum drying could be an alternative?

In my opinion, the most analytical part was done properly. However, there is no subchapter for chemicals used in the analysis with a specific supplier. Also, one query due to the chromatographic analysis of bioactive compounds: was any standard or internal standard used in the methodology, cause the results are presented as it says "profile" but expressed as ug/g of dry matter? Moreover, the Authors assure us in the introduction that glucosinolates are important compounds for broccoli, then there is no chromatographic analysis of these compounds, only TGC. It has lower scientific value for readers. 

I also suggest doing an easy Pearson correlation study of antimicrobial, neuroprotective and antioxidant activity activities with chromatographic analysis, TGC, TFC and TPC. 

Author Response

Reviewer#2

  1. The information in the introduction is really scattered. "Despite broccoli popularity..." There is no consumer data used by the Authors to confirm that statement.

Authors: We are grateful for the helpful comments that reviewer #2 has made. Please find below our itemized responses to your comments.

We agree, the statement "Despite broccoli popularity" has been changed by "To our knowledge". In addition, we added some data about consumption and global production of broccoli in Introduction section.

  1. Line 53: the idea of broccoli powders as a functional food doesn't fit the info of broccoli cultivation and therefore, with the idea of making powders.

Authors: We agree, it has been rephrased the paragraph to avoid confusion.

  1. The aim of this research is not well presented. Furthermore, drying is an expensive method and uses a lot of energy. According to the energy crisis is there any issue that vacuum drying could be an alternative?

Authors: We understand the reviewer's concern. Therefore, we have changed the term "a feasible" by "an alternative" in the following paragraph: "This study will contribute to a better understanding of vacuum drying and provide a feasible method for obtaining broccoli powders of high biological activity". Although VD remains inefficient compared to new drying technologies, still in large scale food industries, less energy efficient drying techniques such as freeze drying and spray drying are used for obtaining of powder vegetables. In this study, we have focused in a real alternative method to large scale food industries, since VD has been a practical method that has been successfully used for many years in several food industries. Instead, majority of the new drying methods are experimental-scale dryers. Therefore, many investigations are required for successful implementation of these methods on an industrial scale for food products with high-quality attributes and energy saving characteristics.

  1. In my opinion, the most analytical part was done properly. However, there is no subchapter for chemicals used in the analysis with a specific supplier.

Authors: We agree, it has been included a subchapter of the chemicals used in the present investigation in the materials and methods section according to the reviewer’s recommendation.

  1. Also, one query due to the chromatographic analysis of bioactive compounds: was any standard or internal standard used in the methodology, cause the results are presented as it says "profile" but expressed as µg/g of dry matter?

Authors: Effectively, we have used the commercially available standards (Sigma). The equations of the calibration curves have been added in the manuscript. The results we expressed as µg/g of dry matter to compare with existent literature on drying of broccoli [Yılmaz et al. (2018) and López-Hernández et al. (2022)].

  1. Moreover, the Authors assure us in the introduction that glucosinolates are important compounds for broccoli, then there is no chromatographic analysis of these compounds, only TGC. It has lower scientific value for readers.

Authors: We consider it to be a valid suggestion. Unfortunately, it was not possible to carry out chromatographic analysis of glucosinolates in this study. However, it is noteworthy that intact glucosinolates are considered biologically inactive compounds, being their breakdown products, especially ITCs, the biologically active compounds. Therefore, a goal for future studies will be to identify and quantify the ITCs from vacuum-dried broccoli as we mentioned in conclusions section. Although the TGC has lower scientific value for readers, with this determination we can elucidate if the enzymatic degradation of glucosinolates was produced after drying to form simultaneously ITCs.

  1. I also suggest doing an easy Pearson correlation study of antimicrobial, neuroprotective and antioxidant activity activities with chromatographic analysis, TGC, TFC and TPC.

Authors: We consider it is a valid suggestion. In the original manuscript, it has already been correlated TPC and TFC with antioxidant activity (DPPH and ORAC) as well as the three main phenolic acids (chlorogenic acid, sinapic acid and caffeic acid) with antioxidant activity. Now, in the revised manuscript, we have correlated TPC, TFC and the three main phenolic acids with their respective neuroprotective activity. However, TGC cannot be correlated with any biological activity (antioxidant, antimicrobial and neuroprotective activities) because intact glucosinolates are considered biologically inactive compounds. On the other hand, as it was stated in the manuscript, antimicrobial activity can be affected by solvent extraction method. Thus, to evaluate antimicrobial effect of our samples, another extraction using pure supercritical CO2 fluid was conducted. This is an appropriate technic for extraction of non-polar compounds and some low molecular weight, volatile, polar compounds. However, it is less effective in the extraction of polar phytochemicals embedded in the cell wall. Therefore, it does not seem appropriate to correlate TPC, TFC and the three main phenolic acids with the antimicrobial activity. We hope you agree.

Round 2

Reviewer 2 Report

In my opinion, the manuscript can be accepted in its present form. All the comments and suggestions were taken into consideration by the Authors.